# A New Algorithm to Estimate Chlorophyll-*A* Concentrations in Turbid Yellow Sea Water Using a Multispectral Sensor in a Low-Altitude Remote Sensing System

**Ji-Yeon Baek** [1], **Young-Heon Jo** [1,*] , **Wonkook Kim** [2] , **Jong-Seok Lee** [1], **Dawoon Jung** [1], **Dae-Won Kim** [1] and **Jungho Nam** [3]

[1] Department of Oceanography, Pusan National University, Busan 46241, Korea; jiyeon@pusan.ac.kr (J.-Y.B.); hot4027@pusan.ac.kr (J.-S.L.); firewall2327@pusan.ac.kr (D.J.); daewon@pusan.ac.kr (D.-W.K.)
[2] Department of Civil and Environmental Engineering, Pusan National University, Busan 46241, Korea; wonkook@pusan.ac.kr
[3] Korea Maritime Institute, Busan 49111, Korea; jhnam@kmi.re.kr
[*] Correspondence: joyoung@pusan.ac.kr; Tel.: +85-51-510-3372

**Abstract:** In this study, a low-altitude remote sensing (LARS) observation system was employed to observe a rapidly changing coastal environment-owed to the regular opening of the sluice gate of the Saemangeum seawall-off the west coast of South Korea. The LARS system uses an unmanned aerial vehicle (UAV), a multispectral camera, a global navigation satellite system (GNSS), and an inertial measurement unit (IMU) module to acquire geometry information. The UAV system can observe the coastal sea surface in two dimensions with high temporal ($1\ \mathrm{s}^{-1}$) and spatial (20 cm) resolutions, which can compensate for the coarse spatial resolution of in-situ measurements and the low temporal resolution of satellite observations. Sky radiance, sea surface radiance, and irradiance were obtained using a multispectral camera attached to the LARS system, and the remote sensing reflectance ($R_{rs}$) was accordingly calculated. In addition, the hyperspectral radiometer and in-situ chlorophyll-*a* concentration (CHL) measurements were obtained from a research vessel to validate the $R_{rs}$ observed using the multispectral camera. Multi-linear regression (MLR) was then applied to derive the relationship between $R_{rs}$ of each wavelength observed using the multispectral sensor on the UAV and the in-situ CHL. As a result of applying MLR, the correlation and root mean square error (RMSE) between the remotely sensed and in-situ CHLs were 0.94 and ~0.8 $\mu$g L$^{-1}$, respectively; these results show a higher correlation coefficient and lower RMSE than those of other, previous studies. The newly derived algorithm for the CHL estimation enables us to survey 2D CHL images at high temporal and spatial resolutions in extremely turbid coastal oceans.

**Keywords:** low-altitude remote sensing system; ocean color; chlorophyll-*a*; multispectral camera; unmanned aerial vehicle

## 1. Introduction

The coastal environment is continuously changing by natural and artificial forces. Accordingly, such environmental changes will have both direct and indirect impacts on the coastal ecosystem and will eventually affect nearby aquafarms and tourism industries. Therefore, it is extremely important to monitor changes to marine environments continuously, and many studies have, therefore, been carried out using various types of observations [1–5]. In general, one of the primary properties used to evaluate changes to a marine environment is the chlorophyll-*a* concentration (CHL) [6–8]. The CHL

has been used as an indicator to identify the biomass of the primary productivity in coastal areas, estuaries, oceanic waters, and lakes [6,9]. It also has been widely used as an indicator of the water quality because it is possible to estimate the algal biomass, which can affect the changes in the marine environment [9].

Methods for measuring the CHL can be divided into direct and indirect observations. Whereas direct observations are a method for collecting water samples from in-situ sites and analyzing water samples [3,8,10], indirect observations refer to methods used to estimate the CHL through the optical water characteristics. Morel and Prierur (1977) [11] used spectrometers to measure the absorption and scattering factors for the seawater characteristics of optics. Later, Mobley (1999) [12] defined the remote sensing reflectance, $R_{rs}$, and effects of physical environmental variables. A variety of ocean color satellite algorithms have been developed that can calculate the CHL using $R_{rs}$. O'Reilly et al. (1998) [13] proposed various algorithms for the detection of the CHL, such as the band-ratio, empirical models, or semi-analytic models; and the color-index based algorithm (CIA) by Hu et al. (2012) [14] was later applied to the Sea-viewing Wide Field-of-view Sensor (SeaWiFS) and Visible Infrared Imaging Radiometer Suite (VIIRS) satellite to estimate the CHL. A study by Blondeau-Patissier (2014) [15] was conducted to examine the limits of the CIA used in a coastal zone color scanner, SeaWiFS, moderate resolution imaging spectroradiometer, and medium resolution imaging spectrometer sensors. However, the CIA is known to have a decreased accuracy in turbid (including the coast) and algae bloom areas.

Ocean color studies based on satellite observations show synoptic and global views of both coastal and open oceans, respectively. The satellite images function as a very useful and effective observation for coastal areas and oceans with low availability of on-site observation data. However, most ocean color satellite data are unavailable in many regions, owing to the heavy clouds covering the oceans. Furthermore, the monitoring of a rapidly changing coastal environment is limited, owing to the low temporal and spatial resolutions. Various attempts have recently been made to overcome the limitations of satellite observations. Among them, studies are being conducted using a UAV, which can be easily applied at a lower cost with breakthrough hardware and software, allowing us to obtain higher spatial resolutions than in satellites images [16–21]. Such an observation system is called a low-altitude remote sensing (LARS) system and collects optical information through sensors mounted on a UAV flying below the clouds. A LARS system is mainly equipped with a digital camera, a multispectral line scan sensor, or a hyperspectral sensor to acquire optical information from the sea surface in two dimensions (2D). Compared to images using a LARS system, which has a spatial resolution of centimeters, the general resolution of ocean color satellite observations is coarse.

Thus, the high-resolution 2D spectral images achieved through LARS allow the monitoring of various small-scale marine phenomena occurring along the coast. Previous studies using LARS have been focused on events such as algae bloom in coastal regions and estuaries. Oh et al. (2016) [20] showed the feasibility of applying the red tide index for red tide bloom occurring near aquafarms using a multispectral camera attached to a UAV. However, it only identified regions with red tide outbreak potential by using k-means clustering based on an arbitrary threshold on the value of the observed water-leaving radiance ($Wm^{-2}\ sr^{-1}\ nm^{-1}$) images in areas where a red tide occurred. Kubiak et al. (2016) [18] analyzed the morphological similarity of a green algae bloom observed through all multispectral data obtained from satellite and LARS images. That study only presented the possibility for identifying visible ocean phenomena, however, rather than providing a quantitative analysis using the radiance of the sea surface. Leighton et al. (2013) [19] also attempted to observe harmful algal blooms (HABs) and thermal plumes on the sea surface using a digital camera and a thermal imaging camera. A morphological analysis of HABs and thermal plumes was conducted using both types of camera, but no quantitative comparative analysis was applied using field observations. In addition, Shang et al. (2017) [21] attached a hyperspectral sensor to a UAV to identify an intense phytoplankton bloom, using $R_{rs}$, as satellite observations were used. The $R_{rs}$ values were calculated using quantitative optical information compared to previous studies, and a fluorescence line height (FLH) algorithm was

applied. However, the point measurement of the hyperspectral sensor has a limitation in creating a 2D image with a high spatial resolution. Therefore, recent studies have attempted to estimate the CHL by acquiring high-resolution images using an UAV. Choo et al. (2018) [17] attempted to measure the CHL quantitatively using a multispectral camera attached to a UAV. A total of 44 images were synthesized, creating an orthophoto around a river. The normalized difference vegetation index (NDVI), which is well known as a vegetated area condition, was calculated using red (680 nm) and near-infrared (840 nm) wavelengths to measure the CHL [22]. The square of the correlation coefficient ($R^2$) of the calculated and in-situ CHLs was 0.7031, indicating a high correlation and the possibility of a quantitative CHL analysis through images captured by a multispectral camera attached to a UAV. However, as Choo et al. (2018) [17] estimated the CHL through the NDVI calculated based on the radiance of the sea surface, rather than $R_{rs}$ observed using a multispectral camera, the calculated data varied according to different light intensities.

Although previous studies using a UAV have attempted to monitor the coastal environment, their validations through in-situ measurements have been limited. In previous UAV studies, only the radiance of the sea surface was used. In that case, the possibility of an error may be increased, depending on the variance in light intensity and the atmospheric conditions. In addition, the wavelength in the most previous studies was limited to a blue-green ratio or NDVI. The limited wavelength makes it extremely difficult to distinguish other suspended matters or a sun glint affecting the CHL [7,23–25]. Thus, we used the $R_{rs}$ of the red edge wavelength for the multi-linear regression (MLR) algorithm in this study, because estimated CHLs have smaller RMSEs compared to in-situ CHLs in Section 4.3. Validity of our study design is supported previously reported studies based on the fact that there are two spectral peaks (around 430 nm and 670 nm) [7,22,26]. It is notable that the red edge wavelength (670 nm) plays an important role in compromising the parameters of MLR for CHL estimation in Section 4.3 [27]. Therefore, this study proposes a new 2D CHL estimation algorithm, applying the LARS system, which can overcome these problems. For the LARS observation, a UAV with a multispectral camera was used. A multispectral camera, RedEdge, can be used as a radiometer and can acquire a total of five wavelength images. We caclulated the radiance for the sea surface obtained through the multispectral camera as $R_{rs}$, similar to the use of $R_{rs}$ for the CIA of a CHL estimation.

In this study, in-situ CHL measurements and an optical survey were conducted on the sea surface around the Saemangeum seawall located off the western coast of South Korea (Figure 1). The radiance and irradiance ($Wm^{-2}\ nm^{-1}$) observed using the multispectral camera were calculated to obtain the values of $R_{rs}$ and were compared with those observed by a hyperspectral radiometer to determine the sensitivity of the sensor.

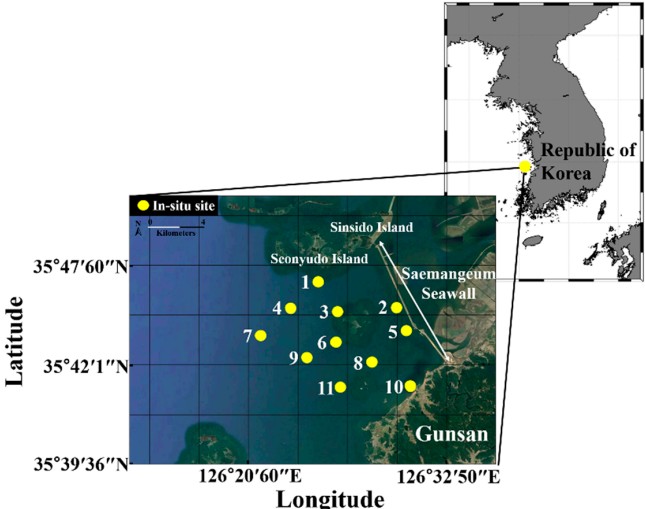

**Figure 1.** Location of the Saemangeum seawall (study area) off the western sea of the Korean peninsula.

In line with the above the purpose of this study was to develop a new CHL estimation algorithm based on multispectral camera signals used to produce 2D CHL data. It is worth noting that the study area is known as an extremely turbid region, thus previous CHL algorithms do not perform well. We, therefore, attempted to develop a better CHL algorithm using a multi-linear regression method in case-II water using a multi-spectral camera on a LARS system, which to the best of the authors' knowledge, is the first time such an approach has been attempted. Section 2 of this paper describes the study area and dataset, and Section 3 details the methods used. The analysis and results of this study are given in Section 4, and some concluding remarks and a discussion are provided in Section 5.

## 2. Study Area and Datasets

### 2.1. Study Area

The study area is the Saemangeum seawall area shown in Figure 1. Saemangeum is located off the western coast of South Korea where a large amount of suspended matter and freshwater flows from three rivers (the Geumgang, Dongjingang, and Mangyeonggang). This region has a shallow water depth (average of ~40 m), thus in winter, the temperature-salinity of the seawater is vertically uniform, owing to the vertical mixing caused by strong seasonal winds and tidal currents. In summer, a strong stratification forms, owing to the release of fresh water and surface heating [28]. Construction of the Saemangeum seawall started in 1991 for land reclamation and securing agricultural water, and was completed in 2010. With the completion of the seawall, which is 33.9 km long, the internal (fresh water) and external (seawater) waters are completely separated and are partially circulated by two sluice gates (Sinsi, Garyeok) located to the south of the seawall.

The sluice gates of the seawall are operated to allow sea water to flow in and fresh water to flow out according to the tidal cycles. When the fresh water released is mixed with seawater, the two different water masses meet, causing ecological and physical changes. Previous studies have shown that the distribution of suspended particle matter (SPM) and the concentration of salinity in the inner (fresh water) and outer (seawater) waters of the seawall are changed drastically through the opening of the sluice gates [3,4,29]. According to Choi (2014) [2], after construction of the Saemangeum seawall was completed, the marine environment outside the seawall became influenced by the fresh water discharged after the opening of the sluice gates. It is known that when the internal water of the seawall, having a relatively high CHL, is released to the outside, seasonal phytoplankton blooming occurs, resulting in an increase in pH and decrease in dissolved oxygen (DO) [2].

Many aquafarms are located off the southern coast of Saemangeum and are directly affected by the fresh water flowing from the two sluice gates. Therefore, the high temporal and spatial-resolution oceanographic observations are needed to monitor these rapidly changing coastal environments. In this study, field and remote sensing observations were conducted off the southern sea area of the Saemangeum seawall, and a total of four in-situ observations were applied (Table 1).

**Table 1.** The in-situ sites used in this study.

| Station | Longitude (° E) | Latitude (° N) | Observation Date (year/month) |
|:---:|:---:|:---:|:---:|
| 1 | 126.420358 | 35.783427 | 2018/4, 5, 10 |
| 2 | 126.499541 | 35.757180 | 2018/4, 5, 10 |
| 3 | 126.439802 | 35.753447 | 2018/4, 10 |
| 4 | 126.392630 | 35.756847 | 2018/4, 5, 8, 10 |
| 5 | 126.509413 | 35.734344 | 2018/4, 5, 10 |
| 6 | 126.438644 | 35.722900 | 2018/4, 5, 8, 10 |
| 7 | 126.362619 | 35.729286 | 2018/4, 5, 8, 10 |
| 8 | 126.474644 | 35.703113 | 2018/4, 8, 10 |
| 9 | 126.409058 | 35.707161 | 2018/4, 5, 10 |
| 10 | 126.514077 | 35.678475 | 2018/4, 5, 10 |
| 11 | 126.442833 | 35.677772 | 2018/4, 5, 8, 10 |

## 2.2. Datasets

### 2.2.1. Low-Altitude Remote Sensing System (LARS)

In this study, a UAV as a type of LARS observation system was used to acquire the remote sensing images. The LARS system is a remote sensing system for operation at an altitude of approximately 500 m to 2 km above the sea surface (applicable when < 500 m). Because observations are conducted under the clouds, cloud cover does not pose a hindrance, and images with high temporal and spatial resolutions can be obtained. In this study, the LARS system was operated using Inspire 2 (Figure 2 and Table 2) of DJI, which is a rotary wing UAV. Inspire 2 can fly at a speed of 26 m s$^{-1}$ for up to 27 min, up to an altitude of 500 m (the flight altitude depends on the local aviation regulations). In addition, it has a global navigation satellite system (GNSS) and an inertial measurement unit (IMU) that can provide the posture and location information of the UAV. Moreover, various sensors, such as digital, multispectral, and thermal-infrared cameras can be attached to the UAV, depending on the purpose of the observations. For this study, a multispectral camera was installed to observe the optical characteristics of the Saemangeum sea surface.

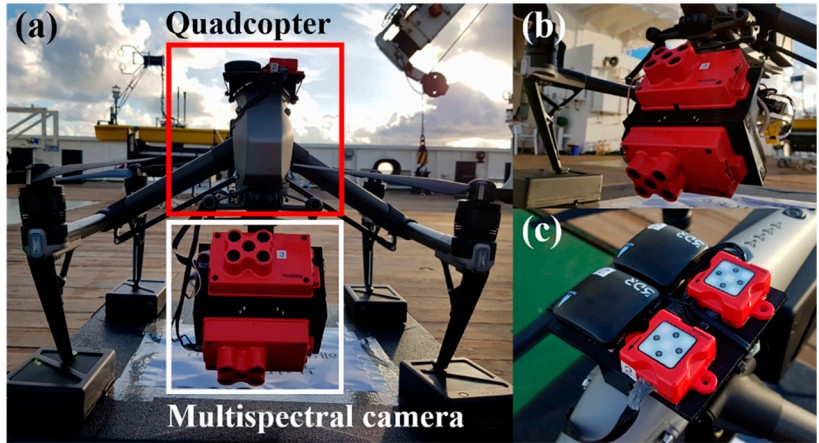

**Figure 2.** Low-altitude remote sensing system (LARS) observation system: (**a**) Inspire 2 (quadcopter); (**b**) multispectral cameras; and (**c**) global navigation satellite system (GNSS) and downwelling light sensor (DLS).

**Table 2.** Inspire 2 (quadcopter) specifications.

| Information | Specification |
| --- | --- |
| Platform | Inspire 2 (quadcopter) |
| Manufacturer | DJI |
| Weight | 3,440 g (include battery) |
| Maximum flight time | ~30 min |
| Maximum flight weight | 4,000 g |
| Maximum flight altitude | 2.5 ~ 5 km |
| Maximum flight speed | 94 km h$^{-1}$ |
| GNSS accuracy | verticality: 0.5–1 m<br>horizontality: 0.3–1.5 m |

### 2.2.2. Multispectral camera: MicaSense RedEdge camera

A multispectral camera (specifically a MicaSense RedEdge camera) has five independent lenses used to capture different wavelengths (Figure 2 and Table 3). In particular, a multispectral camera has radiometric properties that can be used to measure the sea surface reflectance. Therefore, this multispectral camera can be used to calculate the CHL empirically based on the relations between optical properties and in-situ CHL measurements. The camera has a GPS/IMU and records the position

and posture information of the moment of the camera. Moreover, its wireless fidelity (wi-fi) function allows control of the camera settings and checking the observation images in near real-time. In addition, the downwelling light sensor (DLS), which was connected to the camera and installed independently on top of the UAV, can measure the irradiance. The irradiance acquired through the DLS was recorded in the metadata of each camera image and can be used to correct for differences in light intensity owing to the cloud effects or angle of the camera.

**Table 3.** Multispectral camera (RedEdge camera) specifications.

| Information | Specification |
|---|---|
| Platform | RedEdge camera (multispectral camera) |
| Manufacturer | MicaSense |
| Weight | 170 g (includes DLS and cables) |
| Spectral bands (center wavelength, bandwidth (±), nm) | Blue (475, 20), Green (560, 20), Red (668, 10), Red edge (717, 40), Near IR (840, 40) |
| Ground sample distance (GSD) | 8 cm pixel$^{-1}$ (per band) at 120 m |
| Capture rate | 1 capture sec$^{-1}$ (all bands) |
| Field of view (FOV) | 47.2° HFOV |

The multispectral camera can observe five wavelengths ($\lambda$): blue (475 nm), green (560 nm), red (668 nm), red edge (717 nm), and near-infrared (NIR, 840 nm). In general, many algorithms for an ocean color sensor using the blue-to-green band-ratio have been used to calculate the CHL from $R_{rs}$. However, this research used additional wavelengths to increase the accuracy of the chlorophyll-*a* detection. The red (668 nm) and red edge (717 nm) wavelengths were used to identify the properties, presenting a peak in the red wavelength when chlorophyll is combined with other cellular components [7,13,25,30,31]. The NIR wavelength (840 nm) was used to eliminate any sun glint [24].

The multispectral camera was used in two ways depending on its purpose. First, optical information of the sea surface was obtained through the multispectral camera installed on the side of the research vessel at the sea surface and used as input data for the development of the CHL estimation algorithm. According to the method developed by Mobley (1999) [12], the zenith angle ($\theta$) = 40° of the installed multispectral camera was designed to use the sky (sky radiance, $L_{sky}$, Wm$^{-2}$ sr$^{-1}$ nm$^{-1}$) and surface (total upwelling radiance, $L_w$, Wm$^{-2}$ sr$^{-1}$ nm$^{-1}$) directions; and the azimuth angle ($\emptyset$) = 135° was maintained during the observations [24]. A total of 440 datasets were obtained by observing four datasets of images (one set has a total of five images taken by a multispectral camera) with time intervals of 5 s. Second, to obtain 2D sea surface optical information, a multispectral camera was mounted on the UAV to obtain a sea surface image (Figure 2). In this case, the two multispectral cameras on the UAV were designed to view two different angles at $\theta$ = 40° and $\emptyset$ = 135°, which are the same as those installed on research vessels to obtain the imagery at the same time interval [12,24].

### 2.2.3. Hyperspectral Radiometer: TriOS RAMSES

TriOS RAMSES, a hyperspectral radiometer, was used to verify the radiometric calibration for the multispectral camera images (Figure 3). TriOS RAMSES allows optically estimating the components of the water mass by observing the radiometric characteristics of the sea surface. TriOS RAMSES consists of a RAMSES-ACC sensor for irradiance measurements and a RAMSES-ARC sensor for radiance measurements. Here, $L_{sky}$ and $L_w$ were obtained by adjusting the angle of a single RAMSES-ARC. This instrument has a field of view (FOV) of 7° and collects 256 wavelength bands at 3.3 nm intervals within the 320–950 nm range. The RAMSES is mounted on the same frame as the multispectral camera installed on the side of the research vessel. Both sensors measured $L_{sky}$, $L_w$, and the downwelling spectral plane irradiance ($E_d$, Wm$^{-2}$ nm$^{-1}$) at the same time, and the angle ($\theta$ = 40°, $\emptyset$ = 135°) [12]. Observations were conducted on a total of 11 in-situ sites, and each sensor collected ten types of ocean surface optical information at 5 s intervals for each in-situ site. The RAMSES observed data were

interpolated at 1 nm intervals to be compared with the same wavelength as those of the multispectral camera [32].

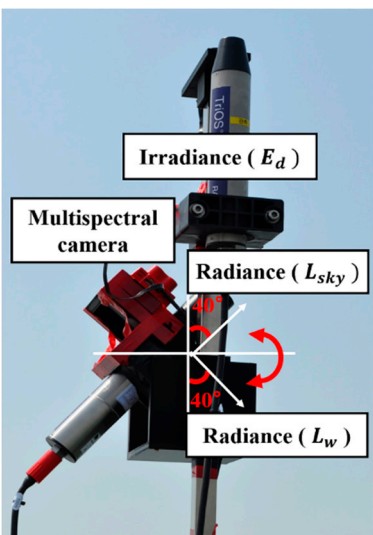

**Figure 3.** RAMSES hyperspectral radiometer and RedEdge multispectral camera. Both sensors measure downwelling spectral plane irradiance ($E_d$), sky radiance ($L_{sky}$) and total upwelling radiance ($L_w$) to estimate reflectance ($R_{rs}$), as discussed in Section 4.2.

### 2.2.4. In-Situ CHL

To verify the CHLs calculated from the optical data of the multispectral camera and hyperspectral radiometer, we measured the CHL using water samples at each in-situ site. The CHL data were obtained four times at the 11 in-situ sites in April, May, August, and October (Table 1). The in-situ observations were conducted by collecting 500 mL of seawater at each site according to the protocols of [8,33]. The sampled waters were filtered using a GF/F filter (47 mm diameter) and placed in a dark room for 24 h after adding 90% acetone. In addition, the CHL was measured using a fluorescence spectrometer (developed by Water and Eco-Bio Co., Ltd.).

### 3. Methods

Figure 4 shows a flowchart of our work flow, which is divided into two processes: one for the development of a CHL estimation algorithm based on the optical data and field observations obtained from the research vessel, and the other for applying the completed algorithm to the multispectral camera images to obtain 2D CHL images.

A linear regression analysis is the most common statistical analysis method used to explain and predict the data. However, if there are multiple independent variables for a phenomenon, a multiple linear regression should be applied, which allows a more accurate and relatively variable estimation [34].

Multi-linear regression (MLR) has been used to study the various complex natural phenomena, as follows. Lee et al. (2017) [35] attempted to estimate the amount of soil moisture through MLR, and Lee et al. (2019) [36] developed an MLR model that predicts the distribution of the soil moisture in South Korea. The basic form of the expression is as follows.

$$Y = \beta_0 + \beta_1 X_1 + \beta_2 X_2 + \cdots \beta_i X_i \tag{1}$$

where Y is the dependent variable; $X_1$, $X_2$, $\cdots$ $X_i$ are independent variables; $\beta_0$ is an intercept; and $\beta_1$, $\beta_2$, $\cdots$ $\beta_i$ are regression coefficients. In this study, MLR was used to determine CHL.

In this study, the relation between each wavelength of the multispectral camera and in-situ CHL was determined through MLR using 120 data points, as described in Section 4.2. Eighty percent of the total data were used to calculate the algorithm, whereas the other 20% of the data were used to verify the established MLR algorithm.

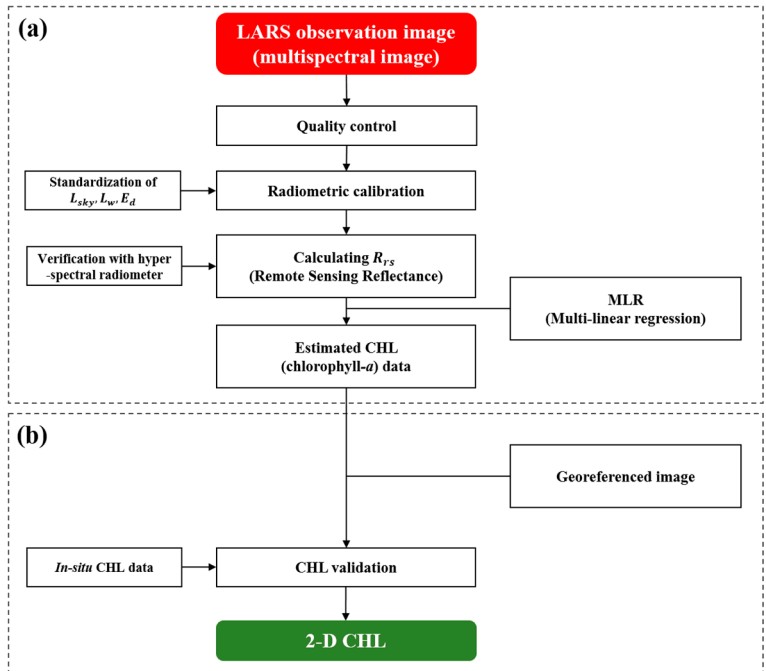

**Figure 4.** Flowchart for (**a**) calculating the chlorophyll-*a* concentration (CHL) and applying the CHL algorithm; (**b**) validation parts of the study.

## 4. Analysis and Results

### 4.1. Radiometric calibration

The optical information obtained by each lens of the multispectral camera is recorded as a digital number (DN). The DN is affected by many factors, such as atmospheric absorption and scattering, and the sensor calibration. Therefore, the multispectral image data must be calibrated to use the actual radiometric characteristics. The DN can be converted into radiances through a radiometric calibration process. For remote sensing observations, the relative radiation calibration (RRN) was employed because in-situ atmospheric data are not required for atmospheric correction. The method includes normalizing or modifying the intensity or DN of each wavelength of a multispectral camera image. Normalized images appear to have been acquired using atmospheric and light conditions similar to those of other images [37]. The radiometric calibration equation (Equation (2)) used in this study is as follows (provided by MicaSense (RedEdge camera company)):

$$L(\lambda) = V_{(x, y)} * a_1/g * \{(P - P_{BL})/t_e + a_2y - a_3t_ey\} \tag{2}$$

where $L(\lambda)$ is the wavelength radiance in $Wm^{-2}\ sr^{-1}\ nm^{-1}$; $V_{(x, y)}$ is the vignette polynomial function for the pixel location $(x, y)$; $a_1$, $a_2$, and $a_3$ are the radiometric calibration coefficients; g is the sensor gain setting; P is the normalized raw DN; $P_{BL}$ is the normalized black level value; and $t_e$ is image exposure time. All values except P are recorded as the metadata. Equation (2) includes a vignette model and pixel value normalization to correct for the lower light sensitivity that occurs in pixels farther from the center of the image. Thus, an image taken of the sky and sea surface direction through Equation (2) is calculated using $L_{sky}$ and $L_w$ (Figure 3). In addition, Ed uses the measurements taken by the DLS sensor. Therefore, the calculated $L_{sky}$, $L_w$, and $E_d$ are used to calculate $R_{rs}$ in the following step.

*4.2. Calculating $R_{rs}$*

Each wavelength $R_{rs}$ (sr$^{-1}$) of the image is calculated using L$_{sky}$, L$_w$, and E$_d$. The $R_{rs}$ equation is as follows:

$$R_{rs}(\lambda) = \{L_w(\theta, \emptyset) - (\rho * L_{sky}(\theta, \emptyset))\}/E_d \qquad (3)$$

Here, ρ is a factor that represents the effects of wind speed, sun glint, and the distribution of the sky's radiance, similar to Fresnel reflectance, where ρ ≈ 0.025 according to Mobley (1999) [12]. The radiance of the image at each wavelength is calculated as $R_{rs}$ using Equation (3). The value of $R_{rs}$ of the multispectral camera (image $R_{rs}$) was verified using the $R_{rs}$ of RAMSES (RAMSES $R_{rs}$) within the same spectral bandwidth. RAMSES $R_{rs}$ is interpolated at intervals of 1 nm for synchronization using the wavelengths of the multispectral camera [32]. The $R_{rs}$ of the quality assurance (QA) system proposed by Wei et al. (2016) [38] defines the $R_{rs}$ spectrum for a total of 23 optical water types and presents the upper and lower bounds for each type of water. Such data were compared with the in-situ data and presented to determine the valid $R_{rs}$ based on a score of between zero and 1. Based on this, we determined the spectrum of water types observed in this study and used only valid data. The RAMSES $R_{rs}$ with a score of over 7/9 from the $R_{rs}$ QA system was used. The image $R_{rs}$ was compared and verified using RAMSES $R_{rs}$ sorted through the QA system. The regression results of the $R_{rs}$ and RAMSES $R_{rs}$ images are shown in Figure 5, and only the data included in the 95% confidence interval were used for analysis. The correlation coefficient (Corr.) of the blue, green, red, red edge, and NIR wavelengths between the two $R_{rs}$ values were 0.92, 0.95, 0.90, 0.82, and 0.78, respectively. The effect of the sun glint can be minimized using the method developed by Mobley (1999) [12] by applying the coefficient ρ of Equation (3); however, the accuracy can be improved further by removing the sun glint and white caps (Figure 6). In general, in the case of clear water, black pixel assumption allows us to ignore the L$_w$ of the NIR by making it zero, implying that we can reduce the influence of the atmospheric scattering, a white cap, and sun glint by subtracting $R_{rs}$ at the NIR wavelength [39]. In the case of turbid water, the relatively longer wavelength (short wave infrared, 1000–3000 nm) can be used [24,32,40]. Therefore, the same method was used to consider the sun glint and white caps as having zero effect in this study. Therefore, the correction of the sun glint and white caps can be applied indirectly by subtracting the offset of the NIR wavelength at all wavelengths of the $R_{rs}$ and RAMSES $R_{rs}$ images [24,41]. It is worth noting that the correlation in Figure 6d is lower (0.484) than other correlation coefficients. The reason is that the spectral signals at 717 nm is relatively close to those of NIR (840 nm). This idea can be found in Figure 5d,e. The data points in the left lower corner (below 0.013 sr$^{-1}$) of figures have the similar patterns, which are widely scattered. It means that the spectral signals at 717 nm have difficulties removing sun glint using NIR in this study. As a result, the CHL estimation algorithm was calculated using a total of four wavelengths (blue, green, red, and red edge bands).

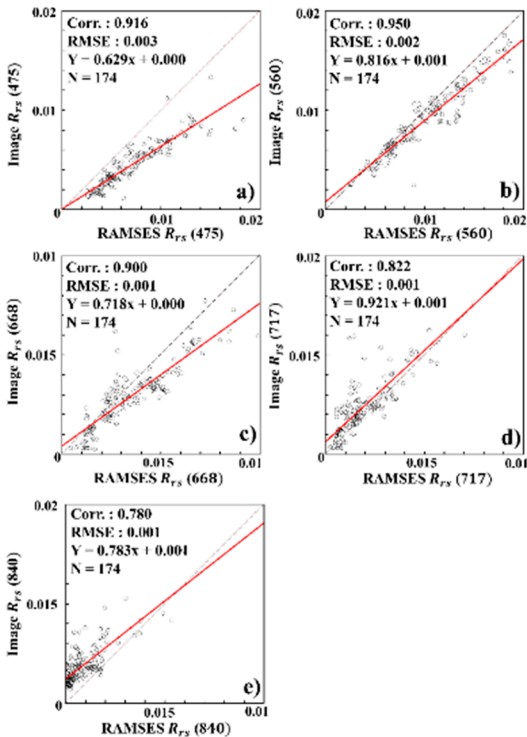

**Figure 5.** Extracted data of more than 95 % significance. Each subfigure is the two different $R_{rs}$ at 475 nm (**a**), 560 nm (**b**), 668 nm (**c**), 717 nm (**d**), and 840 nm (**e**). The x-axis is $R_{rs}$ of the RAMSES (RAMSES $R_{rs}$) and y-axis is $R_{rs}$ of the image (Image $R_{rs}$) value. The black dotted line indicates a one-to-one correspondence and the red solid line indicates the fitting curve between the two data.

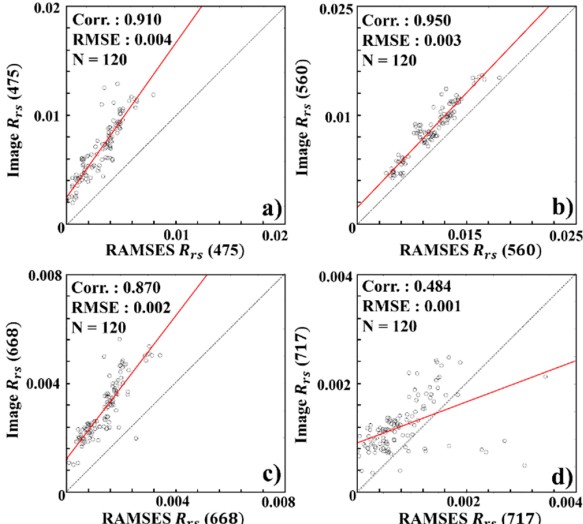

**Figure 6.** The process of eliminating the sun glint from the data. Each subfigure is the two different $R_{rs}$ at 475 nm (**a**), 560 nm (**b**), 668 nm (**c**), and 717 nm (**d**). The x-axis is $R_{rs}$ of the RAMSES (RAMSES $R_{rs}$) and y-axis is $R_{rs}$ of the image (Image $R_{rs}$) value. The black dotted line indicates a one-to-one correspondence and the red solid line indicates the fitting curve between the two data.

*4.3. CHL Algorithm*

The CHL algorithm was developed by applying $R_{rs}$ data from the RAMSES and multispectral cameras to the MLR method described in Section 3. In-situ CHL observations were conducted at the same position (the side of the ship), where the two sensors were aimed up and down, and the image observed with the multispectral camera was extracted in the same area as RAMSES (the FOVs

of RAMSES and RedEdge are 7° and 47.2°, respectively). Because the FOVs of the two sensors are different, a multispectral camera image was cropped and used similar to that of the FOV of RAMSES. The CHL algorithm was determined through MLR as follows:

$$CHL = 9.750 - 391.413\, R_{rs(\lambda\_475)} - 378.371\, R_{rs(\lambda\_560)} - 109.880\, R_{rs(\lambda\_668)} + 158.540\, R_{rs(\lambda\_717)} \quad (4)$$

A total of four wavelengths were used, and the NIR wavelength was not included in Equation (4) because it was assumed to be zero to remove the sun glint and white caps, as described in Section 4.2. The four wavelengths were used to complete the algorithm for estimating the CHL using MLR. A comparison of the results between the MLR-derived CHLs and in-situ CHLs is shown in Figure 7. The correlation between the two data is 0.942, and the root mean square error (RMSE) is approximately 0.8 µg L$^{-1}$ (Figure 7 and Table 4). It is worth noting that we conducted statistical analysis to examine the selection of the regression variables. Different combinations of spectral signals are significant ($p < 0.002$). However, the R$^2$ for MLR with red edge band is higher (R$^2$ = 0.233–0.363) than that without Red edge band (R$^2$ = 0.227–0.240). It seems that the red edge band plays an important role in determination of CHL.

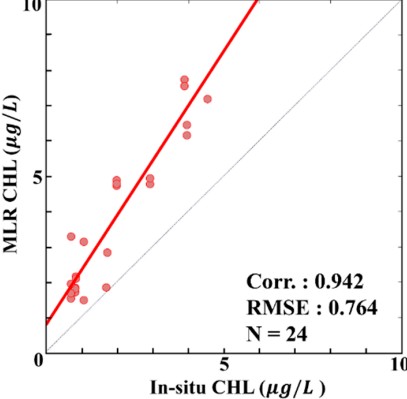

**Figure 7.** Comparison between calculated CHL from multiple linear regression (MLR CHL) and in-situ CHL.

**Table 4.** Correlation between calculated CHL from MLR (MLR CHL) and in-situ CHL.

| Multi linear regression | |
|---|---|
| Equation | CHL = $\beta_0$ + $\beta_1\, R_{rs(\lambda\_475)}$ + $\beta_2\, R_{rs(\lambda\_560)}$ + $\beta_3\, R_{rs(\lambda\_668)}$ + $\beta_4\, R_{rs(\lambda\_717)}$ <br> * $\beta_0$ = 9.750, $\beta_1$ = -391.413, $\beta_2$ = -378.371, $\beta_3$ = -109.880, $\beta_4$ = 158.540. |
| Corr. | 0.942 |

It is worth noting that the SPM concentration ranges of the study area, which has quite turbid water, were approximately 1–34 mg L$^{-1}$ during the research cruises (Table 1). During the observations, the sluice gates of the Saemangeum seawall were opened and SPM and suspended solids accumulated in the fresh water inside the seawall as it was released to the outside. In general, absorption by SPM or colored dissolved organic matter (CDOM) may result in an overestimation of the CHL in marine coastal waters for absorption from a red wavelength, such as with chlorophyll-*a* [7,23]. In addition to the MLR method used to estimate the CHL through Equation (4), the band-ratio method was also examined. Using only the blue and green bands, the estimated CHL shows a correlation of approximately 0.665, and the RMSE was ~1.62 µg L$^{-1}$, although the estimated CHL range was limited to 4–5 µg L$^{-1}$. In addition, for the CHL estimation algorithm without considering the effects of sun glint and white caps, Corr. was ~0.508, and RMSE was ~1.58 µg L$^{-1}$. Therefore, in this study, the MLR method performed better than the conventional approaches.

*4.4. 2D CHL*

　　Equation (4) was applied to the multispectral images obtained by a UAV to produce the 2D CHL on October 25, 2018. The images obtained had a high spatial resolution of $30 \times 50$ cm2 pixel$^{-1}$ at a 500 m altitude. Each image was transformed into a georeferenced image using the actual coordinates by applying direct georeferencing through GPS/IMU data recorded in the metadata of the image [42]. The acquired images were processed as described in Sections 4.1 and 4.2, and the calculated $R_{rs}$ of each wavelength was applied to Equation (4). In Figure 8, the right-hand-side images in gray scale were the original multispectral images for the individual wavelengths, and the left-hand-side images were the 2D CHLs calculated from Equation (4). It is worth noting that one can see aquafarm sites and wave features in high spatial resolution images (Figures 8b–e and 8g–j). We then compared the in-situ CHLs observed at the same time and locations from the CHL images (Figure 8a,f) where the georeferencing was applied [42]. To minimize the effects of image noise, the CHL based on a multispectral camera used a range of data (20 pixels $\times$ 20 pixels) at the location where the in-situ CHL observation was conducted. In order to examine the atmospheric signals below 500 m altitude, we measured irradiance at different heights and found that its effect is very small (0.1 Wm$^{-2}$ nm$^{-1}$ for NIR ~0.13 Wm$^{-2}$ nm$^{-1}$ for red wavelength), causing uncertainties about 9% of $R_{rs}(\lambda)$. Although the atmospheric irradiances are small below 500 m altitudes, atmospheric correct needs to be applied, which will be our next research topic LARS observations. While Figure 8a shows the in-situ CHL of 2.251 µg L$^{-1}$ and MLR derived CHL of 3.895 µg L$^{-1}$, Figure 8f shows the in-situ CHL of 3.591 µg L$^{-1}$ and MLR derived CHL of 3.930 µg L$^{-1}$. The white color in Figure 8a,f show where an invalid $R_{rs}$ did not calculate the CHL. In addition to the changes in the marine environment owing to the opening of the sluice gates, differences in the estimated CHLs could occur because the image can see a broader range than the point data, similar to a hyperspectral radiometer or in-situ data. Therefore, because each pixel has a different wind velocity and solar reflectance, the Fresnel reflectance may vary. However, in this study, the Fresnel reflectance was used at a fixed value of 0.025 under the assumption that the wind velocity and solar reflectance for each pixel are the same. If a valid Fresnel reflectance is used for each image pixel, it is possible that this differences between in-situ and remotely sensed spectral signals are small. In addition, the subtracting of $R_{rs}$ at NIR can further reduce sun glint and atmospheric scattering signals in the imagery [39]. Both approaches were applied in this study, as shown with Figure 8.

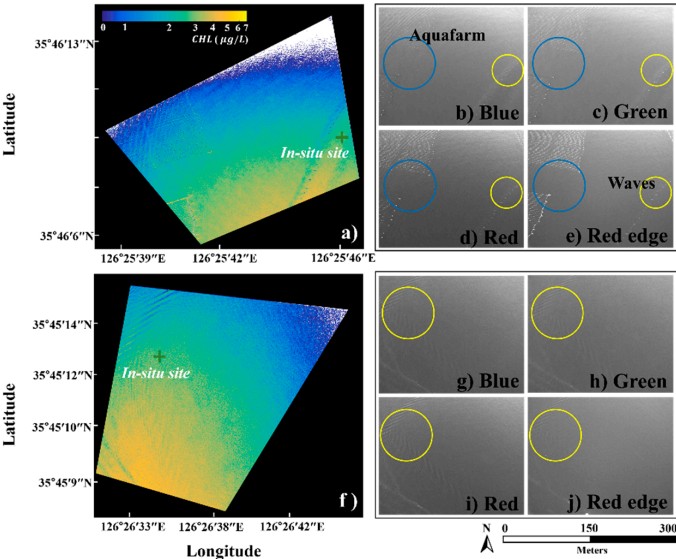

**Figure 8.** 2D CHL (**a**) derived from multispectral camera images (**b–e**) and another 2D CHL (**f**) derived from multispectral camera images (**g–j**) on October 25, 2018. While (**b–e**) and (**g–j**) were not georeferenced, (**a,f**) were georeferenced. The cross marks represent the in-situ site. The blue and yellow circles represent the aquafarm site and the waves features, respectively.

## 5. Discussion and Conclusion

Most of UAV based studies on ocean phenomena have focused on analyzed only the visible features on the sea surface by using typically known methods, such as the band-ratio or CIA algorithm. In terms of advancement of method, we aimed to obtain high temporal and spatial resolution images using a UAV to estimate the 2D CHL through MLR in a quantitative manner. The LARS system can be operated below the clouds to create a 2D CHL with a high temporal and spatial resolution. For a quantitative analysis of the optical information on the sea surface, the spectral image of the sea surface observed through the multispectral camera on the research vessel was converted into the radiance through a radiometric calibration. In addition to blue, green, and red wavelengths, the red edge wavelengths were hired to distinguish the suspended matters, and the NIR wavelength was used to identify the effects of sun glint. The converted radiance of each wavelength was calculated as $R_{rs}$. After the calculation, a valid $R_{rs}$ was extracted according to the method proposed by Wei et al. (2016) [38]. The sun glint and whitecaps were then removed according to the method by Kay et al. (2009) [24], and four wavelengths (blue, green, red, and red edge) were used for the MLR. MLR is a general method for describing the relationship between multiple independent variables through methods [34], and the algorithm has been shown to perform well. Before using MLR, the band-ratio method using blue and green wavelengths was attempted, although the correlation with the in-situ CHLs was ~0.6 and the RMSE was ~1.6 µg L$^{-1}$, and the range of the band-ratio derived CHL values was limited to 4–5 µg L$^{-1}$. Since band-ratio algorithm was based on only two bands to estimate the CHL, it was not better than the MLR method due to the effects of the white cap and sun glint. Therefore, a new 2D CHL estimation algorithm was calculated using the MLR method by adding more spectral signals (especially $R_{rs}$ of the red edge and NIR wavelengths) to reduce the effects of SPM, sun glint, and white caps.

The MLR method was used to determine the relationship between the in-situ CHLs and the four wavelengths of $R_{rs}$. The correlation between the MLR-derived CHLs and in-situ CHLs was 0.942 (R$^2$ of 0.887), indicating an extremely high correlation. In a similar study, Oh et al. (2016) [20] attempted to calculate the red tide index, although a quantitative verification with the measured red tide index was not conducted. In addition, Shang et al. (2017) [21] applied $R_{rs}$ to the FHL algorithm. As a result, the difference between the FHL-derived CHL and in-situ CHL value was less than 20%, although a CHL image with a high spatial resolution could not be generated. Choo et al. (2018) [17] used the NDVI to calculate the CHL calculated in their study.

Next, using a multispectral camera attached to a UAV, images with a high temporal and spatial resolution of 30 × 50 cm$^2$ pixel$^{-1}$ at an altitude of 500 m were obtained, and each image was georeferenced directly [42]. Accordingly, it was possible to calculate the 2D CHL data with the latitude and longitude coordinates per pixel, which enabled a comparison with the in-situ CHLs observed at the same time and location. Comparison of two CHL datums at the same position indicated, that, a 2D CHL was generally ~0.8 µg L$^{-1}$ higher than an in-situ CHL. This may have been influenced by changes in the coastal environment, caused by release of SPM and the suspended solids accumulated in the seawall as the sluice gates opened. This suggests that the calculated CHL is slightly overestimated, as the optical distinction between chlorophyll and turbidity from the increased suspended solids becomes more difficult. In addition, because the lens of the multispectral camera receives the perspective light, the Fresnel reflectance of the 2D image observed at an altitude of approximately 500 m may vary with each pixel. Assuming the Fresnel reflectance is uniform, we witnessed additional overestimation occurred in the UAV campaigns used in this study because it is assumed that the Fresnel reflectance is uniform, so the same value was used for all pixels. If we can apply a different Fresnel reflectance, as calculated by Mobley (1999) [12] to each pixel, we can reduce the error further. Thus, we can confirm that a high-resolution 2D CHL can be obtained in a large area using the algorithm developed in the present study.

In this study, a high-resolution 2D CHL can be estimated through the optical characteristics of the sea surface observed using a LARS system. Furthermore, the estimated CHL algorithm simulates the CHL around the coastal regions well, which is considered meaningful findings compared with

previous studies. The 2D CHL based on a LARS image has an extremely high spatial resolution (~cm) compared to the spatial resolution of an ocean color satellite (~km), allowing for a variety of studies on more marine coastal environments. As shown in the 2D CHL, features such as aquafarms and the movement of ocean currents can be seen in detail. Moreover, ocean phenomena, such as the oceanic front where two water bodies meet, are observed well. In addition, these high-resolution images can contribute to observation of dynamics of small-scale ocean phenomena, which can be expected to help in effectively identifying and analyzing red tide patches or coastal bloom, or tracking changes in the CHL.

**Author Contributions:** Conceptualization, J.-Y.B. and Y.-H.J.; data curation, J.-Y.B., J.-S.L., D.J., and D.-W.K.; investigation, J.-Y.B., J.-S.L., D.J., and D.-W.K.; methodology, J.-Y.B., W.K., Y.-H.J., and J.N.; formal analysis, J.-Y.B., Y.-H.J., and W.K.; writing—original draft, J.-Y.B.; writing—review and editing, Y.-H.J.; supervision, Y.-H.J.

**Funding:** This research was a part of the project "Integrated management of marine environment and ecosystems around Saemangeum," funded by the Ministry of Oceans and Fisheries, Korea (grant number 20140257). We would like to thank Editage (www.editage.co.kr) for English language editing.

**Acknowledgments:** We appreciate four anonymous reviewers for providing useful suggestions and comments.

**Conflicts of Interest:** The authors declare no conflict of interest.

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
