# Peer review of "A New Algorithm to Estimate Chlorophyll-A Concentrations in Turbid Yellow Sea Water Using a Multispectral Sensor in a Low-Altitude Remote Sensing System"

_remotesensing, doi:10.3390/rs11192257_

Round 1
Reviewer 1 Report
General comment:
This paper shows possibility of a low-altitude remote sensing system for the regional environmental monitoring.
Authors say this is a quantitative analysis, however, more consideration , e.g., about the atmospheric correction and regression scheme, seem to be required as the quantitative analysis.
Specific comments:
L70-71:
"Such an observation system is called a low-altitude remote sensing (LARS) system and collects optical information through sensors mounted on a UAV flying below the clouds."
Authors mention the merit of the LARS which can observe below the clouds.
Can you mention about the difference of the optical condition (i.e., the atmospheric correction, not only the irradiance normalization) between clear sky (with direct solar light) and below clouds (only diffused light)?
Can Eq(3) be applied in the cloudy condition?
L84-85:
"This study presented only the possibility of identifying visible ocean phenomena, however, rather than providing a quantitative analysis using the radiance of the sea surface."
The LARS assumes the operation in 500-2000m (mentioned in Line 168) altitudes, which means there are about 5%-20% of the atmospheric molecules below the sensor and the low-altitude aerosols can affect the observed radiance significantly.
If authors aim quantitative analysis of the radiance, the atmospheric correction (consideration of scattering and absorption of the molecules and aerosols) should be mentioned in this paper.
L111-112:
"Thus, we used the red edge wavelength in this study because it can be used to distinguish a suspended matter and the CHL from the observed images."
Authors say the "Red-dedge observation" as a key characteristics of the LARS system.
However, there is no description about how affect the "red edge observation" to the quantitative estimation of the CHL including the discrimination between the suspended matter and CHL in this paper.
L252-253:
"Multi-linear regression (MLR) has been used to study the various complex natural phenomena as follows."
Rrs in the visible-NIR bands have correlation (not independent) among the bands to the CHL values because the absorption and back scattering spectrum of phytoplankton ranges widely in UV to NIR.
L257-258
"In this study, MLR was used to determine how much each wavelength is related to the in-situ CHL."
When there are correlation among the variables, the regression results cannot directly show the correlation of each wavelength with CHL.
If you want to know how much each wavelength Rrs is related to the in-situ CHL, you can use orthogonalization or correlation between each band and the in-situ CHL, I think.
table.3: "170g"
I feel 170g seems too light as the multi band optics.
Please check the specification value.
Fig. 4:
"Flowchart for (a) calculating the CHL and (b) applying the CHL algorithm to the images."
The figure is confusing; it seems that (a) calculating Rrs and applying the CHL algorithm, and (b) validation parts of the study.
Eq.(2):
Why "y" is included in the last two terms (a2y a3ey)?
Eq.(3), L287: "Here, rho is the effect on the sun glint,"
rho seems represent not only the sunglint also reflection of scattering light from the sky.
L303-304:
"In general, for clear water without suspended particles, the effects of a white cap or sun glint may be considered to be zero [11, 32]."
The sentence seems confusing.
Even for clear water without suspended particles, the effects of a white cap or sun glint is not zero.
Authors indicate that the effects of the white cap or remaining sun glint after eq.(3)
can be corrected by subtracting the Rrs at NIR from Rrs at other wavelengths in the case of clear water without suspended particles?
To help readers to understand the process, can you write the equations about the correction by NIR?
The subtraction of Rrs(NIR) seems to be a partial correction of the atmospheric scattering below the sensor.
Even if so, that is not sufficient because the reflectance of the atmospheric scattering (including aerosol scattering) is not constant among the wavelengths).
L323-324:"(the FOVs of RAMSES and RedEdge are 8 and 47.2 degrees, respectively)"
L222:"This instrument has a field of view (FOV) of 7"
Which is correct, 8 degrees or 7 degrees?
L326, Eq.4:
Please describe by Rrs(lambda_*) instead of lambda_* to avoid misreading by readers.
Fig 7:
I could not read that the results are derived by the RAMSES radiometer or the RedEdge imager from the description of the paper.
I suppose the RAMSES was used for the calibration of the RedEdge imager (red lines of Fig. 6), and
the calibrated RedEdge imager data (120 samples) were used to make the coefficients of eq.(4).
Finally, the coefficients were validated by the other 24 samples.
Is that right?
L369-370:
"If a valid Fresnel reflectance is used for each image pixel, it is possible that this error may be reduced."
The subtraction of Rrs at NIR applied in the paper has the role of the correction of the water surface reflection (including the influence of Fresnel reflection)?
L394-397:
"Through band-ratio process was determined that it is difficult to estimate the CHL using only two bands, and it is necessary to consider the effects of the white caps and sun glint. Therefore, a new 2D CHL estimation algorithm was calculated using the MLR method, when considering the effects of SPM, sun glint, and white caps."
I cannot find the evidence of that the difference of the estimation errors from the band ratio and MLR is caused by the effects of SPM, sun glint, and white caps from this paper.
"Choo et al.(2018) [5] used the NDVI, the CHL calculated from their study, and the R2 of the in-situ CHL, and calculated the CHL as 0.703, which was lower than the results of our study. Thus, it was shown that this correlation is higher than in previous studies."
That cannot be concluded if the validation samples are the common, I think.
Can you test the NDVI method using your same validation dataset?
Reviewer 2 Report
This manuscript aimed to obtain high temporal and spatial resolution images to estimate the 2D CHL through MLR in a quantitative manner. A new 2D CHL estimation algorithm was calculated using the MLR method. The results help to understand a rapidly changing coastal environment. Please find some minor issues to be considered below:
Please explain why the correlation coefficient at red edge (717 nm) is only 0.48 as shown in figure 6d. Table 4 was not mentioned in the manuscript and the coefficients listed in table 4 are not consistent with those in function 4.Author Response
Please see the attachment.

Reviewer 3 Report
This study used the images from UAV to estimate the CHL using the MLR technique. The in-situ data was used to make the model and applied to the image. The topic is rather interest but still have some issues could be improve the quality.
What is the new algorithm have used in this study? I just found that, study used the conventional predictive model as MLR whereas I was found many studies in field of remote sensing applied the complex machine learning technique for remote sensing application. So., please provide the carefully details of novel things that you want to claim in the manuscript. Line 19: Should be using GNSS instead of GPS because it may be Galileo, GLONAS, GPS, etc. Line 111-112: “Thus, we used the red edge wavelength in this study because it can be used to distinguish a suspended matter and the CHL from the observed images.” Could you please provide some references where reported the capability of red edge to distinguished CHL from others suspended matters? Especially for extremely turbid area. Line 158 and Figure1: it will be very good of the authors could overlay or pointed to the aqua-farms in the Figure 1. Table 2 GPS accuracy: what is the technique was used? RTK? How the Horizontal accuracy? Line 253-254: rather old references. CHL algorithm and Line 319-329 : How about the relation of the date of image acquisition and the date of collecting ground truth data? it is very importance because you have the multi-date of the in-situ data as showed in the Table 1 and this study use the "high temporal" word of the UAV several place but not show how to use the capability of multi-temporal data to make the good model from multi-date of the in-situ data. In the abstract “The UAV system can observe the coastal sea surface in two dimensions with high temporal (1 s-1) “what you want to communicate with the readers? It is very fast as every second? When we process the UAV images the one of complex is the flight plan and the photometric process. But this study not provide any details of this process. How we can make sure all image from UAV especially in the water could be mosaic well. How about the accuracy and error of the photogrammetric process? In the introduction, this study claimed the new idea to include the red-edge in the model to distinguish between CHL and other suspensions matter. It will be good if the authors could show the MLR model that include and not include the Red edge wavelength in the model. Figure 7: The result from the MLR model have overestimated, especially for the CHL more than 2 ug/L this graph make me unbelievable the RMSE from this graph was low as 0.764. Another one is corr. 0.942 is the correlation of the model (as you show in the table 4) not the correlation of the calculated CHL and in-situ CHL. Please Check.Author Response
Please see the attachment.

Reviewer 4 Report
I recommend to accept the paper in the present form
Round 2
Reviewer 1 Report
Authors have improved the paper, however it may still have some concerns and clerical errors.
Point 2:
I intended to ask in the "Point 2" that applicability of Eq(3)
in both the direct irradiance dominant (clear sky) and diffuse irradiance dominant (cloudy sky) conditions, i.e., BRDF issue.
Some explanation or discussion about that will improve the significance of the paper, but it's not mandatory.
Point 3:
Authors mentioned in lines 173-175, "In this study, an UAV as a type of LARS observation system was used to acquire the remote sensing images. The LARS system is a remote sensing system for operation at an altitude of approximately 500 m to 2 km above the sea surface."
As authors indicated, the atmospheric effect can be small within 500-m altitude (and wavelengths longer than green) generally, however, it can be significant if the scheme applied to 2 km altitude or/and heavy aerosol cases.
I think the limitation of the current scheme (applicable when <500m) should be mentioned around the above sentence to avoid misunderstanding by readers.
Point 4:
Line 115-116: "Thus, the red edge wavelength (670 nm) plays important role in compromising the parameters of MLR for CHL estimation in Section 4.3"
670nm is a red channel and the "rededge wavelength" may be 717nm (as your Table 3).
The red edge also can be useful for Chl estimate in very turbid waters such as in-land lakes, and you can refer some past research papers, I suppose.
Point 5:
Lines 266-267: "MLR was used to determine CHL because each Rrs measured by RedEdge sensor is independent to each other"
Even if the spectral response is not overlapped, they are not "independent" because of the strong correlation among the channel signals.
I think MLR can be used to derive the coefficients of the CHL estimate in this study, however,
MLR cannot be used for evaluation of the channel contribution because of their correlation.
Anyway, the phrase, "independent to each other" should not be used because it is confusing, I think.
The correlation table in the response document is interesting, and it can be included in this paper or use your next study, but it's not mandatory.
Point 6:
Again, even if the spectral response is not overlapped, they are not "independent" in the view of information.
So, the phrase, "independent to each other" should not be used because it is confusing, I think.
Point 7:
OK, I confirmed that it is the very small sensor.
Point 8:
“(a) calculating Rrs and applying the CHL algorithm, and (b) validation parts of the study”seems OK, but
Actually that is as follows in the new document:
"(a) calculating the CHL and applying the CHL algorithm, and (b) validation parts of the study."
Point 9:
OK, I understand that the reason of "y" is out of scope of your paper.
Point 10:
OK, I confirmed the revision.
Point 11:
I understood what authors intended.
I think it is better to add some words as follows:
"implying that we can reduce the influence of the atmospheric scattering, a white cap and sun glint
by subtracting Rrs at the NIR wavelength".
Point 12:
OK, I confirmed the revision.
Point 13:
OK, I confirmed the revision.
Point 14:
OK, I confirmed.
Point 15:
OK, I confirmed the revision.
Point 16:
The comparison should be conducted on NIR subtracted Rrs, Rrs(blue, green, red, and rededge)-Rrs(NIR).
Generally correlation coefficients increase when the number of variable is increased.
I strongly recommend authors to check the selection of the regression variables by statistics ways (e.g., use of determination coefficient adjusted for degrees of freedom"),
because the usefulness of the red edge band seems one of the key results of this paper.
Point 17:
OK, but the sentence seems still remain in Lines 435-437.
Reviewer 3 Report
The revised manuscript have improved. So, it could be accepted to publication.
